# Text Augmented Spatial-aware Zero-shot Referring Image Segmentation

**Yucheng Suo   Linchao Zhu   Yi Yang** [†]

Zhejiang University, Hangzhou, China

{suoych,zhulinchao,yangyics}@zju.edu.cn

## Abstract

In this paper, we study a challenging task of zero-shot referring image segmentation. This task aims to identify the instance mask that is most related to a referring expression **without** training on pixel-level annotations. Previous research takes advantage of pre-trained cross-modal models, e.g., CLIP, to align instance-level masks with referring expressions. Yet, CLIP only considers the global-level alignment of image-text pairs, neglecting fine-grained matching between the referring sentence and local image regions. To address this challenge, we introduce a Text Augmented Spatial-aware (TAS) zero-shot referring image segmentation framework that is training-free and robust to various visual encoders. TAS incorporates a mask proposal network for instance-level mask extraction, a text-augmented visual-text matching score for mining the image-text correlation, and a spatial rectifier for mask post-processing. Notably, the text-augmented visual-text matching score leverages a $P$-score and an $N$-score in addition to the typical visual-text matching score. The $P$-score is utilized to close the visual-text domain gap through a surrogate captioning model, where the score is computed between the surrogate model-generated texts and the referring expression. The $N$-score considers the fine-grained alignment of region-text pairs via negative phrase mining, encouraging the masked image to be repelled from the mined distracting phrases. Extensive experiments are conducted on various datasets, including RefCOCO, RefCOCO+, and RefCOCOg. The proposed method clearly outperforms state-of-the-art zero-shot referring image segmentation methods.

## 1 Introduction

Different from the traditional semantic segmentation tasks that predict masks belonging to predefined categories (Bolya et al., 2019; Strudel et al.,

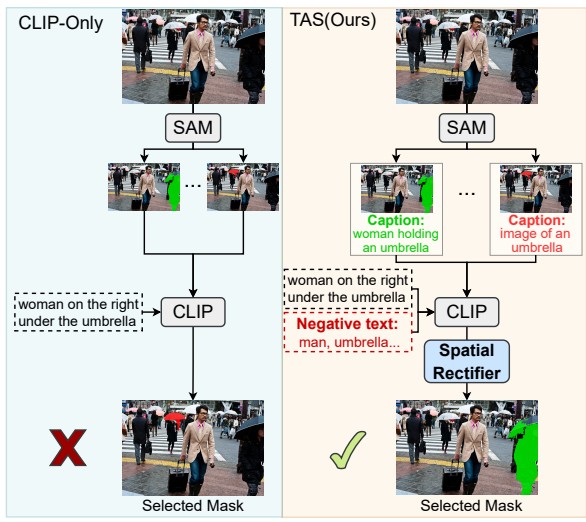

Figure 1: **Main idea of TAS.** CLIP solely calculates the cosine similarity between masked image embedding the text embedding, resulting in picking the wrong mask. TAS uses extra caption embedding, negative text embedding, and a spatial rectifier to rectify the CLIP prediction, thereby picking the correct mask.

2021; Noh et al., 2015; Long et al., 2015), referring expression segmentation is a challenging task that requires identifying a specific object described by a referring expression (Yu et al., 2016; Yang et al., 2022; Wang et al., 2022d; Kim et al., 2022). The task has wide application scenarios such as robot interaction, and image editing (Xu et al., 2023c). The acquisition of precise referring expressions and dense mask annotations is labor-intensive, thereby limiting the practicality in real-world applications. Moreover, the quality and precision of the obtained annotations cannot be guaranteed regarding the labor-intensive annotation process. Therefore, we investigate zero-shot referring image segmentation to reduce labor costs as training on annotations is not required under this setting.

Recently, a zero-shot referring image segmentation framework is proposed (Yu et al., 2023). This framework initially extracts instance masks through an off-the-shelf mask proposal network. Subsequently, the appropriate mask is selected by com-

---

[†]Corresponding author.

puting a global-local CLIP (Radford et al., 2021) similarity between the referring expressions and the masked images. However, the method focuses on the single object in each mask proposal and does not consider other distracting objects within the image. Moreover, since CLIP is trained on image-text pairs, directly applying it to the referring expression segmentation task that requires fine-grained region-text matching could degenerate the matching accuracy (Zhong et al., 2022). Another challenge arises from the domain gap between masked images and natural images (Ding et al., 2022; Xu et al., 2022; Liang et al., 2023), which affects the alignment between masked images and referring expressions.

To this end, we introduce a Text Augmented Spatial-aware (TAS) zero-shot referring expression image segmentation framework composed of a mask proposal network, a text-augmented visual-text matching score, and a spatial rectifier. We utilize the off-the-shell Segment Anything Model (SAM) (Kirillov et al., 2023) as the mask proposal network to obtain high-quality instance-level masks.

To enhance the region-text aligning ability of CLIP and bridge the domain gap between the masked images and the natural images, a text-augmented visual-text matching score consisting of three components is calculated. The first score, called $V$-score, is the masked image text matching score used for measuring the similarity between masked images and referring expressions. The second component is the $P$-score. It bridges the text-visual domain gap by translating masked images into texts. Specifically, a caption is generated for each masked image, followed by calculating its similarity with the referring expression. The inclusion of captions enhances the consistency between the referring expressions and the masked images. To improve fine-grained region-text matching accuracy, we further repel distracting objects in the image by calculating the $N$-score. The $N$-score is the cosine similarity between masked images and negative expressions. We mine these negative expressions by extracting noun phrases from the captions of the input images. The mask that is most related to the referring expression is selected according to a linear combination of the above scores.

Another challenge arises from the limitation of CLIP in comprehending orientation descriptions, as highlighted by (Subramanian et al., 2022). To address this issue, we propose a spatial rectifier as a post-processing module. For instance, to find out the mask corresponding to the referring expression "man to the left", we calculate the center point coordinates of all the masks and pick the mask with the highest text-augmented visual-text matching score from the left half of the masks.

Without modifying the CLIP architecture or further fine-tuning, our method facilitates CLIP prediction using a text-augmenting manner, boosting the zero-shot referring expression segmentation performance. We conduct experiments and ablation studies on RefCOCO, RefCOCO+, and RefCOCOg. The proposed framework outperforms previous methods.

## 2 Related Work

### 2.1 Zero-shot Segmentation

As the rising of image-text pair pre-training multi-modal models like CLIP (Radford et al., 2021), ALIGN (Jia et al., 2021), ALBEF (Li et al., 2021), researchers spend effort in combining cross-modal knowledge (Yang et al., 2021) with dense prediction tasks like detection (Du et al.; Gu et al., 2021; Lin et al., 2022; Bravo et al., 2023; Wang et al., 2023a) and segmentation (Kim et al., 2023; Liang et al., 2022; Luo et al., 2022; Rao et al., 2021; Xu et al., 2023d; Zhou et al., 2021; Qin et al., 2023; Liang et al., 2023; Xu et al., 2023a,b). However, the text used in these works is restricted to object class words or attributes (Li et al., 2022a). Recently, a trend of unified segmentation networks brings dense prediction tasks to a new era (Wang et al., 2023c; Zou et al., 2023). A representative work is Segment Anything Model (SAM) (Kirillov et al., 2023). SAM takes any form of prompt (point, bounding box) to generate masks for a specific area, or to generate masks for all instances without any prompt. A series of works based on SAM aims to apply it in different using scenarios (Ji et al., 2023; He et al., 2023; Li et al., 2023d; Cheng et al., 2023).

### 2.2 Referring Image Segmentation

Referring image segmentation differs from traditional semantic segmentation and instance segmentation since it needs comprehension for a sentence describing a specific object (Hu et al., 2016; Yu et al., 2016; Subramanian et al., 2022). Plenty of fully supervised methods achieve impressive performance (Wang et al., 2022d; Yang et al., 2022;

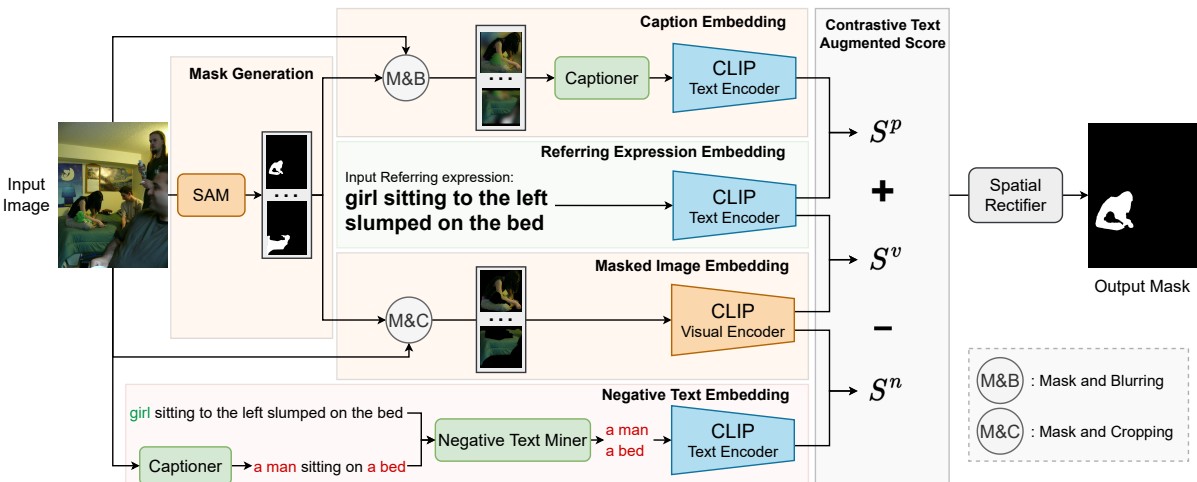

Figure 2: **Overall pipeline of TAS.** Given an input image, we first leverage SAM to obtain instance-level masks. Each mask proposal is applied to the original image to calculate the $V$-score with the referring expression. The captioner generates a caption for each masked image to calculate the $P$-score. Negative expressions are mined to calculate the $N$-score. The mask with the highest score is picked after the post-processing of a spatial rectifier.

Xu et al., 2023c; Liu et al., 2023; Kim et al., 2022; Luo et al., 2020; Ding et al., 2021; Huang et al., 2020), yet these works require pixel-level annotations along with precise referring expressions which are labor-intensive. Recently, a weakly supervised method is proposed, which trains a network only based on the image text pair data (Strudel et al., 2022). Another work goes a step further by utilizing CLIP to directly retrieve FreeSOLO (Wang et al., 2022c) proposed masks without any training procedure (Yu et al., 2023).

### 2.3 Image Captioning

Image captioning, a classic multi-modal task, aims to generate a piece of text for an image(Li et al., 2023b; Changpinyo et al., 2021; Mokady et al., 2021; Wang et al., 2022a, 2023b, 2022a). As the training data amount gets tremendous, the parameter of the state-of-the-art models grows rapidly (Li et al., 2022c; Zhang et al., 2022; Bao et al., 2022; Chung et al., 2022; Wang et al., 2022b; Alayrac et al., 2022). Recent advance in large language models enriches the text diversity of the generated captions (Zhu et al., 2023; Brown et al., 2020; Ouyang et al., 2022). In this paper, we adopt the widely used image captioning network BLIP-2 (Li et al., 2023a).

## 3 Method

### 3.1 Overall Framework

This paper focuses on zero-shot referring expression image segmentation. Our main objective is to enhance the fine-grained region-text matching capability of image-text contrastive models and

bridge the gap between masked images and natural images (Liang et al., 2023) without modifying the model architecture. To achieve the goal, our intuition is to exploit fine-grained regional information using positive and negative texts since text descriptions summarize the key information in masked images. Therefore, we propose a new Text Augmented Spatial-aware (TAS) framework consisting of three main components: a mask proposal network, a text-augmented visual-text matching score, and a spatial rectifier. The mask proposal network first extracts instance-level mask proposals, then the text-augmented visual-text matching score is calculated between all masked images and the referring expression to measure the similarity between masks and the text. After post-processed by the spatial rectifier, the mask most related to the referring expression is selected.

### 3.2 Mask Proposal Network

Previous works (Yu et al., 2023; Zhou et al., 2021) indicate that it is suboptimal to directly apply CLIP on dense-prediction tasks. Therefore, we follow previous works (Yu et al., 2023; Liang et al., 2023; Xu et al., 2022) to decompose the task into two procedures: mask proposal extraction and masked image-text matching. To obtain mask proposals, we adopt the strong off-the-shell mask extractor, i.e., SAM (Kirillov et al., 2023), as the mask proposal network. The mask proposal network plays a vital role as the upper bound performance heavily relies on the quality of the extracted masks.

**FreeSOLO vs. SAM.** Zero-shot referring expression segmentation is to identify a specific object

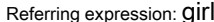

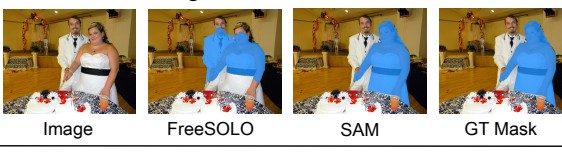

Referring expression: player in red and white

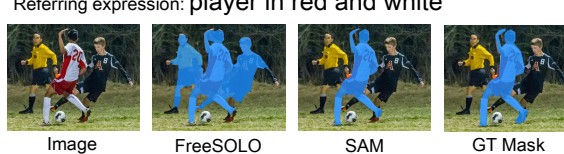

Figure 3: **Comparison between mask proposal networks.** Mask that has the largest IoU with the ground truth mask is visualized.

according to the referring expression. Hence, mask proposal networks with a stronger ability to distinguish instances can yield higher upper-bound performance. Previous work leverage FreeSOLO (Wang et al., 2022c), a class-agnostic instance segmentation network, to obtain all masks. However, we empirically find that the recently proposed SAM (Kirillov et al., 2023) shows strong performance in segmenting single objects. Figure 3 presents a qualitative comparison between mask proposal networks. SAM exhibits superior performance in separating objects, achieving a higher upper-bound performance. We observe that FreeSOLO faces challenges in distinguishing instances under occlusion or in clustered scenarios, whereas SAM is capable of handling such situations effectively. To achieve higher performance, we opt for SAM as the mask proposal network.

### 3.3 Text-augmented visual-text matching score

The mask proposal network provides instance-level masks, but these masks do not inherently contain semantics. To find the mask most related to the referring expression, the typical method is to calculate the cosine similarity between masked images and the referring expression using image-text contrastive pre-trained models like CLIP. One of the issues is that CLIP may be incapable of fine-grained region-text matching (Zhong et al., 2022) since it is trained on image-text pairs. Moreover, the domain gap between masked images and natural images degenerates the masked image-text matching accuracy. To alleviate these issues and facilitate CLIP prediction, we mine regional information using complementary texts. Therefore, we introduce a text-augmented visual-text matching score composed of a $V$-score, a $P$-score, and an $N$-score.

**$V$-score.** Given an input image $I \in \mathbb{R}^{H \times W \times 3}$ and

a referring expression $T_r$. SAM extracts a series of binary masks $\mathbb{M}$ from the image. Every mask proposal $m \in \mathbb{M}$ is applied to the input image $I$. Then the foreground area of the masked image is cropped and fed to the CLIP visual encoder following the approach of previous works (Yu et al., 2023; Liang et al., 2023). The visual feature and text feature extracted by CLIP are used to calculate the cosine similarity. This procedure can be formulated as:

$$I_m = \text{crop}(I, m), \tag{1}$$

$$\mathbf{S}_m^v = \cos(\text{E}_v(I_m), \text{E}_t(T_r)), \tag{2}$$

where crop represents the masking and cropping operation. $E_v$ and $E_t$ indicate the CLIP visual encoder and the CLIP text encoder, respectively. cos means the cosine similarity between two types of features. We term the result as $\mathbf{S}^v$, which represents the visual-text matching score.

Note that the CLIP vision encoder and the CLIP text encoder can be substituted by any image-text contrastive pre-trained models.

**$P$-score.** As mentioned earlier, the domain gap between the natural images and masked images affects the visual-text alignment. To bridge this gap, we introduce a $P$-score to improve the alignment quality by leveraging a surrogate captioning model. The idea is to transfer the masked images into texts, which provides CLIP with complementary object information. Specifically, we use an image captioning model to generate a complementary caption $C$ for each masked image. We encode the captions using the CLIP text encoder and calculate the cosine similarity with the referring expressions. The procedure can be summarized as:

$$\mathbf{S}_m^p = \cos(\text{E}_t(C_m), \text{E}_t(T_r)). \tag{3}$$

$\mathbf{S}^p$ is the $P$-score measuring the similarity between captions and referring expressions. Note that the $P$-score is flexible to any captioning model. However, the effectiveness of $\mathbf{S}^p$ highly depends on the quality of generated captions. Better caption models could bring higher performance.

**$N$-score.** $V$-score and $P$-score promote alignment between masked images and referring expressions. Considering the existence of many objects in the image being unrelated to the referring expression, we further propose an $N$-score to filter out these objects. To identify distracting objects, we collect negative expressions for these objects. Then we regard the similarity between masked images and negative expressions as a negative $N$-score. The

effectiveness of the score depends on these negative expressions. To mine unrelated expressions, we first generate an overall caption for the input image. The overall caption summarizes all objects in the image. We then extract noun phrases from the caption using spacy (Honnibal and Johnson, 2015) and regard them as potential negative expressions. Note that there might be phrases indicating the same object in the referring expression. To avoid this situation, we use Wordnet (Miller, 1995) to eliminate the phrases that contain synonyms with the subject in the referring expression. Specifically, we calculate the path similarity of the two synsets to determine whether to eliminate the synonyms. Empirically, we find the strict rules help TAS to identify distinct objects in these datasets. For instance, we believe "young man" and "the boy" are not synonyms. This ensures that the negative objects identified are distinct from the object mentioned in the referring expression. The remaining noun phrases set $\mathbb{T}_n$ is used to calculate the cosine similarity with the masked images. $\mathbf{S}^n$ is defined as the averaged similarity value over the phrases:

$$\mathbf{S}_m^n = -\frac{1}{|\mathbb{T}_n|} \sum_{T \in \mathbb{T}_n} \cos(\mathrm{E}_v(I_m), \mathrm{E}_t(T)). \quad (4)$$

It is worth mentioning that $\mathbf{S}^n$ is a negative score since it measures the probability of a masked image representing an object unrelated to the target referring expression. We enhance fine-grained object region-text matching by eliminating regions for distracting objects. $\mathbf{S}^n$ is also flexible to captioning models, while detailed captions help to capture more negative expressions.

**The text-augmented visual-text matching score.** The final text-augmented visual-text matching score can be obtained by linearly combining the three above-mentioned scores since all scores are the cosine similarity calculated in the common CLIP feature space. The output mask is the one with the highest score.

$$\mathbf{S}_m = \mathbf{S}_m^v + \alpha \mathbf{S}_m^p + \lambda \mathbf{S}_m^n, \quad (5)$$

$$\hat{m} = \underset{m \in \mathbb{M}}{\operatorname{argmax}} \mathbf{S}_m. \quad (6)$$

The final mask $\hat{m}$ is selected by choosing the one with the highest $\mathbf{S}$. Without changing the feature space and modifying the structure, the text-augmented visual-text matching score enhances fin-grained region-text matching only using augmented texts.

## 3.4 Spatial Rectifier

As revealed in (Subramanian et al., 2022), the text-image pair training scheme does not consider spatial relations. In other words, CLIP cannot distinguish orientation descriptions such as "the left cat" or "the giraffe to the right". To this end, we propose a rule-based spatial resolver for post-processing forcing the framework to select masks from the specific region. The procedure can be decomposed into three steps: orientation description identification, position calculation, and spatial rectifying.

**Orientation description identification.** First, we extract descriptive words for the subject of the referring expression $T_r$ via spacy (Honnibal and Johnson, 2015) and check whether there are orientation words like "up, bottom, left, right". If no orientation words are found in the descriptive words, we do not apply spatial rectification.

**Position calculation.** Second, to spatially rectify the predictions, we need the location information of each mask proposal. The center point of each mask is used as a proxy for location. Specifically, the center point location of each mask is calculated by averaging the coordinates of all foreground pixels.

**Spatial rectifying.** After obtaining center point locations, we choose the mask with the highest overall score $S$ under the corresponding area of the orientation. For instance, we pick the mask for the expression "the left cat" from the masks whose center point location is in the left half of all the center points. Having this post-processing procedure, we restrict CLIP to pay attention to specific areas when dealing with orientation descriptions, thereby rectifying wrong predictions.

## 4 Experiments

### 4.1 Dataset and Metrics

The proposed method is evaluated on the widely used referring image segmentation datasets, i.e. RefCOCO, RefCOCO+, and RefCOCOg. All images in the three datasets come from the MSCOCO dataset and are labeled with carefully designed referring expressions for instances. We also report the performance on the PhraseCut test set. In terms of the metrics, we adopt overall Intersection over Union (oIoU), mean Intersection over Union (mIoU) following previous works.

| Metric | Visual Encoder | Method | RefCOCO | | | RefCOCO+ | | | RefCOCOg | | |
|---|---|---|---|---|---|---|---|---|---|---|---|
| | | | Val | TestA | TestB | Val | TestA | TestB | Val(U) | Test(U) | Test(G) |
| oIoU | - | Text-only | 18.65 | 17.60 | 18.32 | 21.78 | 21.68 | 20.58 | 25.11 | 25.48 | 25.93 |
| | ResNet-50 | CLIP Surgery | 18.04 | 14.74 | 21.28 | 18.39 | 14.34 | 22.98 | 20.44 | 21.80 | 21.23 |
| | | Score Map | 20.18 | 20.52 | 21.30 | 22.06 | 22.43 | 24.61 | 23.05 | 23.41 | 23.69 |
| | | GradCAM | 23.44 | 23.91 | 21.60 | 26.67 | 27.20 | 24.84 | 23.00 | 23.91 | 23.57 |
| | | Global-Local$^\dagger$ | 24.58 | 23.38 | 24.35 | 25.87 | 24.61 | 25.61 | 30.07 | 29.83 | 29.45 |
| | | Global-Local | 20.54 | 22.15 | 20.35 | 24.41 | 25.73 | 24.58 | 26.75 | 28.86 | 29.01 |
| | | CLIP-only | 23.41 | 24.77 | 24.07 | 29.83 | 35.46 | 26.25 | 31.26 | 32.47 | 33.08 |
| | | TAS (Ours) | **29.75** | **32.85** | **28.23** | **33.58** | **40.65** | **27.78** | **36.80** | **37.06** | **38.03** |
| | ViT-B/32 | Region Token | 11.56 | 12.37 | 11.35 | 12.54 | 14.07 | 12.22 | 10.87 | 11.51 | 11.74 |
| | | Global-Local$^\dagger$ | 24.88 | 23.61 | 24.66 | 26.16 | 24.90 | 25.83 | 31.11 | 30.96 | 30.69 |
| | | Global-Local | 22.43 | 24.66 | 21.27 | 26.35 | 30.80 | 22.65 | 27.57 | 27.87 | 27.80 |
| | | CLIP-only | 23.01 | 23.18 | 24.27 | 30.05 | 33.74 | 27.01 | 30.66 | 31.41 | 32.27 |
| | | TAS (Ours) | **29.53** | **30.26** | **28.24** | **33.21** | **38.77** | **28.01** | **35.84** | **36.16** | **36.36** |
| mIoU | ViT-S/16 | TSEG | 25.95 | - | - | 22.62 | - | - | 23.41 | - | - |
| | - | Text-only | 26.17 | 24.19 | 25.98 | 30.00 | 29.66 | 29.61 | 35.83 | 35.74 | 36.21 |
| | ResNet-50 | CLIP Surgery | 25.03 | 22.71 | 27.12 | 26.01 | 22.20 | 30.18 | 29.26 | 30.07 | 29.43 |
| | | Score Map | 25.62 | 26.66 | 25.17 | 27.49 | 28.49 | 30.47 | 30.13 | 30.15 | 31.10 |
| | | GradCAM | 30.22 | 31.90 | 27.17 | 33.96 | 25.66 | 32.29 | 33.05 | 32.50 | 33.25 |
| | | Global-Local$^\dagger$ | 26.70 | 24.99 | 26.48 | 28.22 | 26.54 | 27.86 | 33.02 | 33.12 | 32.79 |
| | | Global-Local | 32.73 | 35.31 | 30.09 | 37.74 | 40.69 | 34.93 | 41.62 | 42.88 | 43.96 |
| | | CLIP-only | 34.16 | 35.80 | 31.70 | 41.07 | 46.61 | 34.50 | 43.73 | 44.09 | 45.08 |
| | | TAS (Ours) | **39.91** | **42.85** | **35.85** | **43.99** | **50.58** | **36.44** | **47.68** | **47.41** | **48.69** |
| | ViT-B/32 | Region Token | 17.06 | 18.02 | 16.28 | 18.83 | 20.31 | 17.78 | 16.33 | 16.88 | 17.31 |
| | | Global-Local$^\dagger$ | 26.20 | 24.94 | 26.56 | 27.80 | 25.64 | 27.84 | 33.52 | 33.67 | 33.61 |
| | | Global-Local | 32.93 | 34.93 | 30.09 | 38.37 | 42.05 | 32.65 | 42.02 | 42.02 | 42.67 |
| | | CLIP-only | 33.01 | 33.60 | 31.46 | 40.59 | 43.73 | 35.25 | 42.66 | 42.59 | 44.37 |
| | | TAS (Ours) | **39.84** | **41.08** | **36.24** | **43.63** | **49.13** | **36.54** | **46.62** | **46.80** | **48.05** |

Table 1: Performance on the RefCOCOg, RefCOCO+ and RefCOCO datasets. U and G denote the UMD and Google partition of the RefCOCOg dataset respectively. TAS outperforms all baseline methods in terms of both oIoU and mIoU on different CLIP visual backbones. Note that all methods use mask proposals provided by SAM. † indicates using FreeSOLO to extract masks.

## 4.2 Implementation Details

We adopt the default ViT-H SAM, the hyperparameter "predicted iou threshold" and "stability score threshold" are set to 0.7, "points per side" is set to 8. For BLIP-2, we adopt the smallest OPT-2.7b model. As for CLIP, we use RN50 and ViT-B/32 models with an input size of $224 \times 224$. We set $\lambda$ to 0.7 for RefCOCO and RefCOCO+, 1 for RefCOCOg, and $\alpha = 0.1$ for all datasets. When using SAM as the mask generator, we noticed a high number of redundant proposals. SAM often produces masks for different parts of the same object, leading to issues with visual semantic granularity. In our approach, we have implemented a straightforward yet effective method to filter out irrelevant mask proposals, thereby reducing semantic ambiguity. Specifically, after generating all initial mask proposals, we measure the overlap between any two masks and eliminate smaller masks that are subsets of larger ones. It is important to note that in the experiments section, all methods are evaluated based on these refined mask proposals.

## 4.3 Baselines

Baseline methods can be summarized into two types: activation map-based and similarity-based. For activation map-based methods, we apply the mask proposals to the activation map, then choose the mask with the largest average activation score. Following previous work, Grad-CAM (Selvaraju et al.), Score Map (Zhou et al., 2021), and Clip-Surgery (Li et al., 2023c) are adopted. Note that Score Map is acquired by MaskCLIP. Similarity-based methods are to calculate masked image-text similarities. Following previous work, we adopt Region Token (Li et al., 2022b) which utilizes mask proposals to filter the region tokens in every layer of the CLIP visual encoder, Global-Local (Yu et al., 2023) uses Freesolo as the mask proposal network and calculates the Global-Local image and text similarity using CLIP. Note that for a fair comparison, we also report the results using SAM. Text-only (Li et al., 2023a) is to calculate the cosine similarity between the captions for masked images and the referring expressions. This baseline is to test the relevance of the caption and referring expression.

| Method | Visual Encoder | PhraseCut oIoU | PhraseCut mIoU |
|---|---|---|---|
| Global-Local | - | 23.64 | - |
| TAS(Ours) | ViT-B/32 | 25.64 | 24.66 |
| | ResNet-50 | 25.00 | 24.51 |

Table 2: oIoU and mIoU results on PhraseCut dataset.

| $\alpha$ | RefCOCO oIoU | RefCOCO mIoU | $\lambda$ | RefCOCO oIoU | RefCOCO mIoU |
|---|---|---|---|---|---|
| 0 | 27.41 | 37.89 | 0 | 27.46 | 38.54 |
| 0.05 | 28.75 | 39.35 | 0.3 | 28.62 | 39.56 |
| 0.1 | **29.53** | **39.84** | 0.5 | 29.23 | **39.97** |
| 0.2 | 29.17 | 39.13 | 0.7 | **29.53** | 39.84 |
| 0.4 | 26.93 | 36.39 | 0.9 | 28.96 | 39.18 |

Table 3: Ablation study on the impact of $\lambda$ and $\alpha$. We separately change the value of $\alpha$ and $\lambda$ and test the oIoU and mIoU on the RefCOCO validation set. ViT-B/32 CLIP visual backbone is adopted here.

| Modules $S^p$ | $S^n$ | Spatial | RefCOCO oIoU | RefCOCO mIoU |
|---|---|---|---|---|
| | | | 23.01 | 33.01 |
| ✓ | | | 25.44 | 35.94 |
| ✓ | ✓ | | 27.11 | 37.03 |
| ✓ | ✓ | ✓ | **29.53** | **39.84** |

Table 4: Ablation study on the performance improvement of each proposed module. We report oIoU and mIoU on the validation set of RefCOCO using ViT-B/32 CLIP visual backbone.

CLIP-only (Radford et al., 2021) is a simple baseline that directly calculates the similarity between the cropped masked image and referring expression. We also compare with TSEG (Strudel et al., 2022), a weakly supervised training method.

### 4.4 Results

**Performance on different datasets.** Results on RefCOCO, RefCOCO+ and RefCOCOg are shown in Table 1. For a fair comparison, we reimplement the Global-Local (Yu et al., 2023) method using masks extracted from SAM. TAS outperforms all baseline methods in terms of oIoU and mIoU. Previous works that leverage CLIP visual encoder activation maps perform poorly in all datasets. Compared with the previous SOTA method using FreeSOLO to extract masks, TAS surpasses in both metrics, especially in mIoU. We also report mIoU and oIoU results on the test set of the PhraseCut dataset in Table 2. Our method also outperforms the previous method.

**Qualitative analysis.** Figure 4 shows the qualitative comparison of TAS and previous methods. Note that all the masks are extracted by SAM. TAS is able to comprehend long complex referring expressions and pick the most accurate object mask. With the help of the spatial rectifier, TAS deals well with orientation-related referring expressions.

### 4.5 Ablation Study

**Sensitive toward $\alpha$ and $\beta$.** We propose the text-augmented visual-text matching score, a linear combination of different types of scores. To explore whether the score is sensitive toward the weights $\alpha$ and $\lambda$, we conduct an ablation study. Results are shown in Table 3, $\alpha$ and $\lambda$ are tuned separately. TAS is not sensitive to $\lambda$, we select 0.7 to achieve a balance of mIoU and oIoU improvement. A large $\alpha$ harms the performance, therefore we set the value to 0.1.

**Importance of the proposed modules.** To further prove the effectiveness of the proposed text-augmented visual-text matching score and the spatial rectifier, we conduct an ablation study on the validation set of RefCOCO. The mIoU and oIoU results are reported with different combinations of the modules in Table 4. $S^{cap}$ and $S^{neg}$ are the $P$-score and the $N$-score respectively. "Spatial" represents the spatial rectifier aforementioned. The first line in the table is the result that only uses $S^{img}$, which is also the CLIP-Only baseline result. From the table, we observe that all modules contribute to performance improvement. In particular, the Spatial rectifier plays a vital role in the RefCOCO dataset since RefCOCO contains many orientation descriptions.

**Influence on the input format of masked images.** In table 5, we study two input types of masked images for the BLIP-2 and CLIP. The first method is cropping, which is widely used in previous works(Xu et al., 2022; Liang et al., 2023; Yu et al., 2023). Another method is blurring (Subramanian et al., 2022), we blur the background of the cropped area using a Gaussian kernel. Blurring make the model recognize the mask area with background information. From the table, we find that for the captioning model BLIP-2, blurring is better than crop. However, cropping is better than blurring for CLIP. We suppose the reason is the cropping left black background which helps CLIP to focus on the foreground object. However, for BLIP-2, blurring helps generate context-aware de-

Referring Expression: **kid streaching or whatever he is doing**

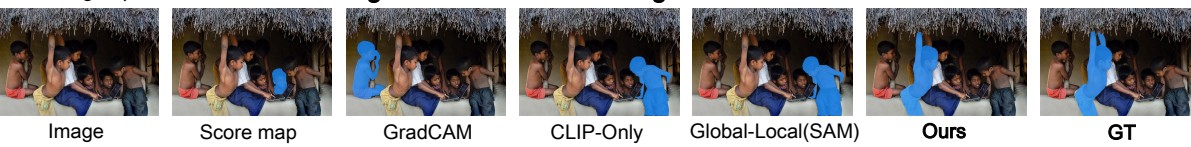

| Image | Score map | GradCAM | CLIP-Only | Global-Local(SAM) | **Ours** | GT |

Referring Expression: **left man jumping to catch frisbee**

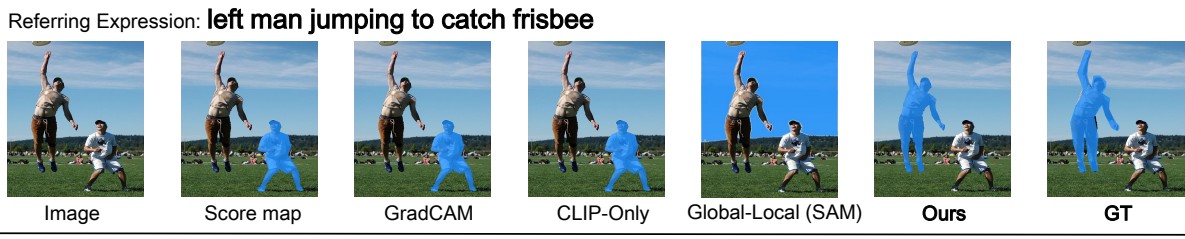

| Image | Score map | GradCAM | CLIP-Only | Global-Local (SAM) | **Ours** | GT |

Referring Expression: **a wooden chair sitting in the corner next to a few rolls of toilet paper**

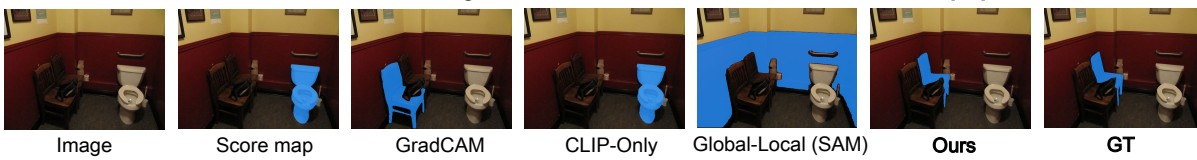

| Image | Score map | GradCAM | CLIP-Only | Global-Local (SAM) | **Ours** | GT |

Figure 4: **Qualitative results of different methods.** All methods are compared using SAM mask proposals.

| Mask Input Type | | RefCOCO | |
|---|---|---|---|
| BLIP-2 | CLIP | oIoU | mIoU |
| crop | crop | 28.87 | 39.08 |
| blur | blur | 27.17 | 37.85 |
| crop | blur | 27.07 | 37.43 |
| blur | crop | **29.53** | **39.84** |

Table 5: Ablation study on the input type of masked images. oIoU and mIoU results are reported on the validation split of RefCOCO using ViT-B/32 CLIP visual backbone.

| Model | Visual Encoder | RefCOCO+ | | RefCOCOg | |
|---|---|---|---|---|---|
| | | oIoU | mIoU | oIoU | mIoU |
| - | ResNet-50 | 35.46 | 46.61 | 31.26 | 43.73 |
| | VIT-B32 | 33.74 | 43.73 | 30.66 | 46.62 |
| BLIP-2 | ResNet-50 | 40.65 | 50.58 | 36.80 | 47.68 |
| | VIT-B32 | 38.77 | 49.13 | 35.84 | 46.62 |
| GIT | ResNet-50 | 40.10 | 50.55 | 35.43 | 46.79 |
| | VIT-B32 | 38.09 | 48.42 | 34.99 | 46.02 |

Table 6: Results on GIT-base image captioning model. We report oIoU and mIoU on the testA split of Ref-COCO+ and RefCOCOg UMD validation split using two kinds of CLIP visual backbone.

scriptions, enhancing comprehension of the referring expression.

**Importance of the image captioning model.** Our intuition is to use texts to enhance region-text alignment and bridge the domain gap between natural images and masked images. The quality of texts depends on the captioning model. To explore the importance of the captioning model, we substitute the BLIP-2 model with GIT-base captioning model (Wang et al., 2022a) and test the performance. Re-

| Method | RefCOCO+ | | RefCOCOg | |
|---|---|---|---|---|
| | oIoU | mIoU | oIoU | mIoU |
| CLIP | 33.74 | 43.73 | 30.66 | 42.66 |
| CLIP-TAS | **38.77** | **49.13** | **35.84** | **46.62** |
| BLIP-2-ITC | 38.78 | 51.75 | 32.99 | 46.91 |
| BLIP-2-TAS | **43.48** | **54.90** | **37.97** | **50.07** |
| Albef-ITC | 7.22 | 11.70 | 10.55 | 15.35 |
| Albef-TAS | **24.59** | **35.10** | **26.58** | **37.20** |

Table 7: Ablation study on image text contrastive model. We report oIoU and mIoU on the testA split of Ref-COCO+ and RefCOCOg UMD validation split using Albef and BLIP-2 image-text contrastive model.

sults are shown in Table 6, we find that the captioning model has little affection on performance. Better captioning models bring better mIoU and oIoU performance.

**Is TAS generalizable to other image-text contrastive models?** To explore whether TAS is generalizable to other image-text contrastive models, we conduct an ablation study, and the results are shown in Table 7. On BLIP-2 (Li et al., 2023a) and ALBEF (Li et al., 2021), TAS makes impressive improvements. We believe TAS is a versatile method that helps any image-text contrastive model.

**Is TAS practicable in real-world scenarios?** TAS does not require high computing resources. All experiments were conducted on a single RTX 3090, which is applicable in real-world applications. The GPU memory consumption for the entire pipeline is about 22GB, including mask generation (SAM), captioner (BLIP2), and masked image-text matching (CLIP). We also test the inference speed on a

random selection of 500 images on a single RTX 3090. The CLIP-Only baseline method (mask generation + masked image-text matching) obtains 1.88 seconds per image. The Global-Local method costs 2.01 seconds per image. Our method TAS (mask generation + captioner + masked image-text matching) achieves 3.63 seconds per image. By employing strategies like 8-bit blip-2 models and FastSAM (Zhao et al., 2023), it would be possible to enhance the efficiency under constrained computational resources.

## 5 Conclusion

In this paper, we propose a Text Augmented Spatial-aware (TAS) framework for zero-shot referring image segmentation composed of a mask proposal network, a text-augmented visual-text matching score, and a spatial rectifier. We leverage off-the-shell SAM to obtain instance-level masks. Then the text-augmented visual-text matching score is calculated to select the mask corresponding to the referring expression. The score uses positive text and negative text to bridge the visual-text domain gap and enhance fine-grained region-text alignment with the help of a caption model. Followed by the post-processing operation in the spatial rectifier, TAS is able to deal with long sentences with orientation descriptions. Experiments on RefCOCO, RefCOCO+, and RefCOCOg demonstrate the effectiveness of our method. Future work may need to enhance comprehension of hard expressions over non-salient instances in the image. One potential way is to leverage the reasoning ability of Large language models like GPT4.

## 6 Limitations

While our approach yields favorable results across all datasets based on mIoU and oIoU metrics, there exist certain limitations that warrant further investigation. One such limitation is that SAM occasionally fails to generate ideal mask proposals, thereby restricting the potential for optimal performance. Additionally, the effectiveness of our approach is contingent upon the image-text contrastive model employed. Specifically, we have found that the BLIP-2 image-text contrastive model outperforms CLIP, whereas the Albef image-text contrastive model shows poor performance when applied to masked images.

Another potential limitation of TAS is the ability to deal with complex scenarios. A potential

research topic is to directly identify the most appropriate mask from noisy proposals. In other words, future works may work on designing a more robust method to deal with the semantic granularity of the mask proposals. Recent work uses diffusion models as a condition to work on this problem (Ni et al., 2023). Finally, the understanding of the metaphor and antonomasia within the referring expression remains insufficient. We observe there are expressions requiring human-level comprehension which is extremely hard for current image-text models. Future work may benefit from the comprehension and reasoning ability of Large Language Models (LLM).

## 7 Acknowledgement

This work is partially supported by Major program of the National Natural Science Foundation of China (Grant Number: T2293723). This work is also partially supported by the Fundamental Research Funds for the Central Universities (Grant Number: 226-2023-00126, 226-2022-00051).

## 8 Ethics Statement

The datasets used in this work are publicly available.

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
