# OpenReview forum: "Text Augmented Spatial Aware Zero-shot Referring Image Segmentation"
_EMNLP/2023/Conference — EMNLP 2023 Findings_

### Official Review · Reviewer_xQwB · 2023-08-01

**Soundness:** 3

**Excitement:**

2: Mediocre: This paper makes marginal contributions (vs non-contemporaneous work), so I would rather not see it in the conference.

**Paper Topic And Main Contributions:**

The author proposes a Referring Expression Segmentation method based on SAM+CLIP+BLIP2, using CLIP for matching and BLIP2 for filtering out irrelevant content in the image.

**Questions For The Authors:**

1. Given the significant differences in the RefCOCO, RefCOCO+, and RefCOCOg datasets, conducting ablation experiments solely on RefCOCO may not fully demonstrate the effectiveness of each module. It would be beneficial to show the performance on other datasets as presented in Table 4.

2. The generated results from the Caption Model often differ greatly from the human-generated Reference. Moreover, images can be described in various ways, and the Negative Text Miner should handle this information correctly. Additionally, the reviewer raises concerns about how the Negative Text Miner deals with phrases in Reference and Caption that have different synonymous angles, such as "a cute girl" and "young lady."

**Reasons To Accept:**

In general, I really like this work because the task itself is valuable. Considering the lack of data for Referring Expression Segmentation, zero-shot methods have always been worth exploring.

However, some major concerns remain.

**Reasons To Reject:**

1. Both BLIP2 and GIT are very large models, which makes this method less practical. Especially for BLIP2, it is even challenging to directly run it on a V100 GPU. Therefore, if we use both SAM and BLIP2, only GPUs with capabilities like A100 or above might be able to handle this method. Additionally, the inference time will be very long, which further limits the real-world application of the method.

2. The method presented in this article is quite similar to the Global-local approach. We can observe that the main difference between this article and Global-local lies in how they handle the local region. S^v is essentially identical to the original article, while S^p in the original article uses a mask on the visual encoding features, whereas this article uses a captioner. As for S^n, which is the main innovation in this article, I consider it not novel enough as similar methods are widely used in multimodal pre-training. Therefore, the contributions of this method seem somewhat incremental.

I may raise the score if the above issues are reasonably addressed.

**Reproducibility:**

4: Could mostly reproduce the results, but there may be some variation because of sample variance or minor variations in their interpretation of the protocol or method.

**Reviewer Confidence:**

4: Quite sure. I tried to check the important points carefully. It's unlikely, though conceivable, that I missed something that should affect my ratings.

---

> ### Author Rebuttal · Authors · 2023-08-29
>
> We appreciate the reviewer's insightful feedback. Here are our responses to your concerns.
> ***
> ### The reviewer mentioned that BLIP2 and GIT are very large models, which makes this method less practical.
>
> **Response:** Thank you for asking about computing resource requirements. It is indeed an important aspect of whether the method can be conducted in real-world scenarios.
>
> **For hardware requirements**, all experiments were conducted on a single RTX 3090. The entire pipeline (SAM+BLIP2+CLIP) consumes about 22GB GPU memory. It is feasible in real-world scenarios.
>
> **In terms of Inference Latency:**, we assessed inference speed using 500 random images on an RTX 3090. The CLIP-Only method (SAM+CLIP), results in 1.88 seconds per image. The Global-Local method (SAM+CLIP) costs 2.01 seconds per image. TAS (SAM+BLIP2+CLIP) achieves 3.63 seconds per image.
>
> To save GPU memory and increase inference speed on a fixed dataset, we only run the captioning model once and store the generated captions for later use. Using the stored captions, the memory consumption reduces to 11GB. This is feasible even on an RTX 2080Ti. The inference time for a single image also decreases to 1.958 seconds.
>
> Overall, our method does not require high computing resources. Even though incorporating a caption model increases inference time, we obtain improvement in both oIoU and mIoU metrics, e.g., +8.15 oIoU, +7.05 mIoU on RefCOCO TestA set. Optimization methods like model quantization may further boost inference speed, we leave it as future work.
> ***
> ### The reviewer mentioned that the method presented in this article is quite similar to the Global-local approach.
>
> **Response:** Our method is different from previous methods. The main contribution of our work is designing a text-driven framework for better masked image-text matching.
>
> The reviewer commented the $S_v$ was identical to previous works. Indeed, $S_v$ is widely used in previous works related to open vocabulary segmentation [1][2][3][4]. We clarified in our submission that this operation follows previous works (Line 269, Line 270).
>
> $S_p$ is different from the masked visual embedding introduced in the Global-Local paper.
> In the Global-Local paper, the authors use the mask on the visual features to obtain global visual features. In contrast, we introduced a text-text matching score ($S_p$) based on the auxiliary description of each masked image for better region-text matching. This paradigm has not been studied in previous works related to zero-shot referring segmentation. Furthermore, the qualitative experiments in Figure 4 showed that masked visual embedding used in the Global-Local method favors larger masks, while our method is able to locate small objects.
>
> $S_n$ is novel in zero-shot referring segmentation. Motivated by repelling unrelated objects, we propose a negative text miner to extract unrelated expressions, enhancing the prediction by an elimination process. We first generate an overall caption for each input image. The overall caption summarizes all objects in the image. We then extract noun phrases from the caption using spacy and regard them as potential negative expressions. The similarity between masked images and negative expressions is termed as a negative score $S_n$. $S_n$ measures the probability of an object being unrelated to the referring expression. Results in  Table 4 demonstrate the effectiveness of $S_n$, e.g. +1.67 oIoU and +1.09 mIoU on RefCOCO.
>
> In essence, our method is different from the Global-Local method in several aspects:
> 1. **Flexibility with Model Architecture:**
> TAS is flexible with image-text matching models and captioning models. Table 6 and Table 7 in the paper show the versatility. In Table 6, using captions generated by the GIT-base model, TAS can still obtain +4.35 oIoU and +4.69 mIoU improvement on the RefCOCO+ TestA set. In Table 7, we substitude the image-text contrastive model CLIP by ALBEF and BLIP-2 and still observe a performance boost. Especially, we observed a surge of +17.37 oIoU and +23.4 mIoU on the ALBEF model.
> 2.  **Integration of Text-Text Matching:**
> We improved the typical masked image-text matching by incorporating descriptions of masked regions. We term the text-text matching score as $S_p$. Having $S_p$, we attain performance with respect to oIoU and mIoU. For instance, $S_p$ brings +2.43 oIoU and +2.93 mIoU in Table 4.
> 3. **Text-Centric Object Modeling:**
> TAS identifies unrelated image objects exclusively via negative text mining. The intuition is to filter distracting objects using negative expressions. The negative expressions are mined from the caption of the input images. We name the similarity between masked images and negative expressions $S_n$. As mentioned earlier, $S_n$ brings performance improvement as shown in Table 4.
> 4. **TAS is spatial aware:**
> TAS is able to deal with spatial descriptions as we designed a spatial rectifier. The performance boost in Table 4 demonstrates the effectiveness.
>
> ***
> Our responses to the questions from the reviewer are listed below:
> ### Question 1. The reviewer asks about the ablation study on different datasets.
> **Response:** Thank you for the question. We have provided the ablation studies on the RefCOCO+ and RefCOCOg in the table below. Note that for the visual encoder, we adopt the VIT-B/32 CLIP backbone.
>
> In terms of the RefCOCO+ TestA set:
> |   $S_p$  |   $S_n$  |  Spatial |   oIoU   |   mIoU   |
> | :------: | :------: | :------: | :------: | :------: |
> |          |          |          |  33.74   |   43.73  |
> |  &check; |          |          |  36.72   |   47.25  |
> |  &check; |  &check; |          |  38.50   |   48.85  |
> |  &check; |  &check; |  &check; |  38.77   |   49.13  |
>
> In terms of the RefCOCOg Umd validation set:
> |   $S_p$  |   $S_n$  |  Spatial |   oIoU   |   mIoU   |
> | :------: | :------: | :------: | :------: | :------: |
> |          |          |          |   30.66  |   42.66  |
> |  &check; |          |          |   32.41  |   44.50  |
> |  &check; |  &check; |          |   35.38  |   46.16  |
> |  &check; |  &check; |  &check; |   35.84  |   46.62  |
>
> We will add the ablation results in the final revision. Our results demonstrate the effectiveness of the proposed method. We observe the major improvement brought by $S_p$ and $S_n$, which validates our motivation of using text for better region-text alignment.
>
> ### Question 2. The reviewer asked about how the Negative Text Miner deals with phrases in Reference and Caption that have different synonymous angles.
> **Response:** As mentioned in the $S_n$ section in the paper, we utilize Wordnet to judge whether the subject noun of the referring expression and the noun of each phrase extracted from the caption are synonyms. Specifically, we calculate the path similarity of the two synsets to determine whether to eliminate the synonyms. Empirically, we find the strict rules help TAS to identify distinct objects in these datasets. For instance, we believe "young man" and "the boy" are not synonyms. This involves the problem of semantic granularity, future work may investigate the influence of it.
>
> ***
> ### Further discussion on our motivation and idea
> We are grateful the reviewer likes the task of zero-shot referring segmentation. Our intuition is to use text to enhance the region-text matching in a zero-shot manner. As the growth in works related to multi-modal areas, it is promising to directly integrate pretraining models[5][6] to solve different tasks. Considering the potential assistance between modalities[7][8], we investigate a text-driven manner for the zero-shot referring segmentation task without training or fine-tuning. Our method obtains state-of-the-art performance on the zero-shot referring expression segmentation task.
>
> In summary, we hope our response addresses the concerns. These inquiries indeed mirror the challenges encountered during our research. We hope for a reconsideration of our submission. We believe the contribution our our paper matches the theme of the track.
> ***
> ### References
> [1]Xu, Mengde, et al. "A simple baseline for open-vocabulary semantic segmentation with pre-trained vision-language model." European Conference on Computer Vision. Cham: Springer Nature Switzerland, 2022.
>
> [2]Subramanian, Sanjay, et al. "Reclip: A strong zero-shot baseline for referring expression comprehension." arXiv preprint arXiv:2204.05991 (2022).
>
> [3]Jiang, Kenan, et al. "Comclip: Training-free compositional image and text matching." arXiv preprint arXiv:2211.13854 (2022).
>
> [4]Liang, Feng, et al. "Open-vocabulary semantic segmentation with mask-adapted clip." Proceedings of the IEEE/CVF Conference on Computer Vision and Pattern Recognition. 2023.
>
> [5]Zeng, Andy, et al. "Socratic models: Composing zero-shot multimodal reasoning with language." arXiv preprint arXiv:2204.00598 (2022).
>
> [6]Shen, Yongliang, et al. "Hugginggpt: Solving ai tasks with chatgpt and its friends in huggingface." arXiv preprint arXiv:2303.17580 (2023).
>
> [7]Lin, Zhiqiu, et al. "Multimodality helps unimodality: Cross-modal few-shot learning with multimodal models." Proceedings of the IEEE/CVF Conference on Computer Vision and Pattern Recognition. 2023.
>
> [8]Radford, Alec, et al. "Learning transferable visual models from natural language supervision." International conference on machine learning. PMLR, 2021.

---

### Official Review · Reviewer_deYa · 2023-08-04

**Soundness:** 4

**Excitement:**

4: Strong: This paper deepens the understanding of some phenomenon or lowers the barriers to an existing research direction.

**Paper Topic And Main Contributions:**

The paper discusses the task of zero-shot referring image segmentation, which aims to identify the instance mask that is most related to a referring expression without training on pixel-level annotations. The authors propose a Text Augmented Spatial-aware (TAS) framework that incorporates a mask proposal network, a text-augmented visual-text matching score, and a spatial rectifier for mask post-processing. The framework leverages a P-score and an N-score in addition to the visual-text matching score to improve alignment between referring sentences and local image regions. The proposed method outperforms state-of-the-art zero-shot referring image segmentation methods in experiments conducted on various datasets.

**Questions For The Authors:**

Question A: While the performance gains on the Refcoco testB dataset are noteworthy, could you provide further results from other datasets to help validate the method's effectiveness and generalizability?
Question B: In terms of transformer-based segmentation methods, particularly the recently popular Meta-SAM, could you outline the advantages of the proposed approach? It would be beneficial to highlight the unique strengths of your method.
Question C: Visualization of instances where the method does not perform as expected might offer a valuable perspective on the current limitations. It could also help in identifying potential areas for improvement.
Question D: Some parts of the paper could benefit from further refinement to enhance the readability and logical flow. For instance, it might be helpful to consolidate some sections in the methods part.
I hope these queries will aid the authors in enhancing the strength of the paper.

**Reasons To Accept:**

1. Novelty: The paper proposes a new Text Augmented Spatial-aware (TAS) framework for zero-shot referring expression image segmentation. The framework consists of three main components: a mask proposal network, a text-augmented visual-text matching score, and a spatial rectifier. This approach introduces a novel way to leverage text information for image segmentation tasks.

2. Performance Improvement: The paper demonstrates that the TAS framework outperforms previous methods in zero-shot referring expression segmentation. The proposed framework achieves higher accuracy and handles challenging scenarios such as occlusion and clustered objects effectively.

3. Ablation Study: The paper includes an ablation study to analyze the effectiveness of each proposed module in the TAS framework. The results show that all modules contribute to performance improvement, and the spatial rectifier plays a vital role in handling orientation-related referring expressions.

4. Experimental Evaluation: The paper conducts experiments and evaluation on multiple datasets, including RefCOCO, RefCOCO+, and RefCOCOg. The results demonstrate the superiority of the TAS framework over previous methods in terms of segmentation accuracy.

Overall, the paper presents a framework for zero-shot referring expression image segmentation and provides thorough experimental evaluation and analysis. The results demonstrate the effectiveness of the proposed approach and its superiority over previous methods. Therefore, I recommend weakly accepting this paper.

**Reasons To Reject:**

The proposed method is valuable and can effectively improve the performance of this task, but the authors use the Segmentation At Large Model (SAM) as priori knowledge model. Is this fair compared to other training methods (e.g., the weakly supervised training method TSEG) in the experiment?

**Reproducibility:**

4: Could mostly reproduce the results, but there may be some variation because of sample variance or minor variations in their interpretation of the protocol or method.

**Reviewer Confidence:**

3: Pretty sure, but there's a chance I missed something. Although I have a good feel for this area in general, I did not carefully check the paper's details, e.g., the math, experimental design, or novelty.

**Typos Grammar Style And Presentation Improvements:**

Line 78: Rephrase "its defects also obvious" to "its limitations are also apparent" for better clarity.
Line 61: Add "the" before "early fusion framework".
Line 139: Change "Endow the network" to "Endows the network" to fix grammar error.
Line 316: Simplify phrase "obtained before" to just "obtained".

---

> ### Author Rebuttal · Authors · 2023-08-29
>
> We thank the reviewer for your precious comments. We are very grateful that the reviewer appreciates our work. Our responses to the raised concerns are below.
> ***
> ### The reviewer is concerned about whether the comparison to other training methods (e.g., the weakly supervised training method TSEG) in the experiment is fair.
>
> **Response:** Thank you for the question. In the experiment section, we compared both weakly supervised methods (e.g., TSEG) and zero-shot methods (e.g., CLIP Surgey, Global-Local, CLIP-only) to demonstrate the effectiveness of our approach.
>
> Compared to weakly supervised training methods (e.g. TSEG), our method is more feasible and effective in real-world applications since it is training-free. Note that the settings of TSEG and our approach are different. TSEG is a weakly supervised method trained on the image-text pairs in the training set of RefCOCO, RefCOCO+, and RefCOCOg. It leverages the activation map toward expressions as the dense prediction results. **In contrast**, our approach follows the zero-shot setting, which does not require human-described referring expressions and is flexible with different off-the-shell models. Notably, our method outperforms TSEG even though no training data is leveraged. In addition, our method leverages SAM [1] for mask generation and SAM is trained on the SA1B dataset without accessing any COCO images. This indicates that TAS follows the zero-shot setting and it is capable of performing tasks without prior training or collecting specific examples. It is worth mentioning that the previous zero-shot method [2] also compared the weakly supervised method TSEG. We followed their setting and reported the result of TSEG as well.
>
> In the revision, we will clearly specify the training setting of the methods (e.g., weakly-supervised or zero-shot).
>
> ***
> ### The reviewer asked about the performance gain in other datasets.
> **Response:** Thank you for the question. We have provided the ablation studies on the RefCOCO+ and RefCOCOg datasets. We adopted the VIT-B/32 CLIP backbone in the ablation studies.
>
> In terms of the RefCOCO+ TestA set:
> |   $S_p$  |   $S_n$  |  Spatial |   oIoU   |   mIoU   |
> | :------: | :------: | :------: | :------: | :------: |
> |          |          |          |  33.74   |   43.73  |
> |  &check; |          |          |  36.72   |   47.25  |
> |  &check; |  &check; |          |  38.50   |   48.85  |
> |  &check; |  &check; |  &check; |  38.77   |   49.13  |
>
> In terms of the RefCOCOg Umd validation set:
> |   $S_p$  |   $S_n$  |  Spatial |   oIoU   |   mIoU   |
> | :------: | :------: | :------: | :------: | :------: |
> |          |          |          |   30.66  |   42.66  |
> |  &check; |          |          |   32.41  |   44.50  |
> |  &check; |  &check; |          |   35.38  |   46.16  |
> |  &check; |  &check; |  &check; |   35.84  |   46.62  |
>
> These results demonstrate the effectiveness of each proposed module. We hope that our response comprehensively addresses the concerns articulated by the reviewer. Thank you for pointing out the typos in our paper.
> ***
> ### References
> [1] Kirillov, Alexander, et al. "Segment anything." arXiv preprint arXiv:2304.02643 (2023).
>
> [2] Yu, Seonghoon, Paul Hongsuck Seo, and Jeany Son. "Zero-shot Referring Image Segmentation with Global-Local Context Features." Proceedings of the IEEE/CVF Conference on Computer Vision and Pattern Recognition. 2023.

---

### Official Review · Reviewer_VMqU · 2023-08-05

**Soundness:** 4

**Excitement:**

3: Ambivalent: It has merits (e.g., it reports state-of-the-art results, the idea is nice), but there are key weaknesses (e.g., it describes incremental work), and it can significantly benefit from another round of revision. However, I won't object to accepting it if my co-reviewers champion it.

**Missing References:**

N/A

**Paper Topic And Main Contributions:**

This work leverage off-the-shelf SAM model and pretrained CLIP to handle zero-shot referring expression segmentation.
Specifically:
- Clever combination of negative phrase together with S score (visual text matching) and P score (alignment quality from captioning model)
- Strong zero-shot performance on widely-used benchmarks

**Questions For The Authors:**

- This approach uses large models such as SAM, BLIP2 and CLIP. This might limit the applicability in the real-world use case. In this era where models pretrained on large-scale datasets are dominant, I am personally fine with that. But please mention your hardware configuration so that readers can correctly assess this method.

**Reasons To Accept:**

- Performance is great! Nice work
- Extensive ablation study and well-executed experiments

**Reasons To Reject:**

- As this approach combines multiple models and requires (1) mining negative phrases (2) spatial rectifier, I wonder what is the inference cost compared to the baseline, as latency is also important.

**Reproducibility:**

3: Could reproduce the results with some difficulty. The settings of parameters are underspecified or subjectively determined; the training/evaluation data are not widely available.

**Reviewer Confidence:**

2: Willing to defend my evaluation, but it is fairly likely that I missed some details, didn't understand some central points, or can't be sure about the novelty of the work.

**Typos Grammar Style And Presentation Improvements:**

Minor point but you should use \citet instead of \cite when the citation is used as a textual object, e.g. line 357

---

> ### Author Rebuttal · Authors · 2023-08-29
>
> We would like to thank you for your precious questions. Here are our responses to your questions.
> ****
> ### The reviewer asked about hardware requirements and the inference latency.
>
> **Response:** Thank you for asking about computing resource requirements.
>
> **Hardware requirements**: Our method does not require high computing resources. Under the zero-shot setting, we do not finetune or train the models. All experiments were conducted on a single RTX 3090, which is applicable in real-world applications. The GPU memory consumption for the entire pipeline is about 22GB, including mask generation (SAM), captioner (BLIP2), and masked image-text matching (CLIP).
>
> **Inference latency:** We have tested the inference speed on a random selection of 500 images on a single RTX 3090. The CLIP-Only baseline method (mask generation + masked image-text matching)  obtains 1.88 seconds per image. The Global-Local method costs 2.01 seconds per image. Our method TAS (mask generation + captioner + masked image-text matching) achieves 3.63 seconds per image.
>
> Given a fixed set of images to be referred to, TAS can be further accelerated by preprocessing image captions beforehand. By using the preprocessed captions, our TAS achieves 1.96 seconds per image and the GPU memory cost (mask generation + masked image-text matching) reduces to 11GB. Although introducing a captioning model inevitably extends inference time, it significantly improves the performance under the mIoU and oIoU metrics, e.g., +8.15 on oIoU and +7.05 on mIoU on the RefCOCO TestA set. The inference speed can be improved by using model quantization and other techniques. We will leave this as the future work of our work.
>
> We will add a detailed inference speed comparison in the revision.
>
> ***
> We are grateful that the reviewer appreciates our work. Thank you for pointing out our typos. We will revise the writing carefully. We are happy to have more discussions about the paper if you have any further questions.

---

### Meta-Review · Area_Chair_jE7G · 2023-09-23

**Recommendation:** 3

**Metareview:**

The paper presents a novel zero-shot referring expression segmentation approach utilizing the SAM model, pretrained CLIP, and BLIP2. Pros include: (1) Strong performance on widely-used benchmarks, as highlighted by its zero-shot capabilities and innovative combination of visual-text matching and alignment quality scores. (2) Comprehensive ablation studies and well-conducted experiments that contribute to a deeper understanding of the method's efficacy. (3) A unique Text Augmented Spatial-aware (TAS) framework that offers new avenues for leveraging text information in image segmentation tasks.
Cons comprise: (1) A sense of incrementalism, with some components of the proposed method bearing similarity to existing methods, thus questioning its true novelty. (2) Lack of clarity on the real-world applicability of the Negative Text Miner in handling diverse image descriptions and semantically varied phrases. After rebuttal and discussion, the AC feels that most concerns are addressed and this work is solid enough to appear in the conference but reviewers are generally less excited about it for the novelty concern.

---

### Decision · Program_Chairs · 2023-10-07

**Decision:**

Accept-Findings

**Comment:**

The paper presents a novel zero-shot referring expression segmentation approach utilizing the SAM model, pretrained CLIP, and BLIP2. Pros include: (1) Strong performance on widely-used benchmarks, as highlighted by its zero-shot capabilities and innovative combination of visual-text matching and alignment quality scores. (2) Comprehensive ablation studies and well-conducted experiments that contribute to a deeper understanding of the method's efficacy. (3) A unique Text Augmented Spatial-aware (TAS) framework that offers new avenues for leveraging text information in image segmentation tasks.
Cons comprise: (1) A sense of incrementalism, with some components of the proposed method bearing similarity to existing methods, thus questioning its true novelty. (2) Lack of clarity on the real-world applicability of the Negative Text Miner in handling diverse image descriptions and semantically varied phrases. After rebuttal and discussion, the AC feels that most concerns are addressed and this work is solid enough to appear in the conference but reviewers are generally less excited about it for the novelty concern.